# The Effect of Social Capital on Auditor's Performance

Maryamalsadat Mousavi Azghandi [1], Sahar Jabbari [2,*], Hossien Rezaei Ranjbar [3] and Ahmed Al-janabi [2]

[1] Department of Economics and Administrative Sciences, Imammareza International University of Mashhad, Mashhad 9177948974, Iran
[2] Department of Economics and Administrative Sciences, Ferdowsi University of Mashhad, Mashhad 9177948974, Iran
[3] Department of Economics and Administrative Sciences, Qaenat Branch, Islamic Azad University, Qaenat 9147921074, Iran
[*] Correspondence: saharjabbari26@yahoo.com

**Abstract:** This paper investigates the relationship between social capital and auditor's performance in Iranian listed firms. The sample included 128 firms on the Tehran Stock Exchange from 2014 to 2020. The research method was descriptive-correlational, and the relationship between research variables was explained using regression models based on the panel data. The results illustrated that social capital positively correlates with auditor performance and Audit report quality. In other words, social capital decreases audit opinion shopping, audit expectation gap, internal control weakness, and audit report lag. Therefore, society's influential assets, social capital, and audit report quality strongly influence the auditor's performance. The auditor's performance affects the probability of discovery and reporting material errors and misstatements. Therefore, recognizing influential factors on auditors' performance can improve the quality of audit reports.

**Keywords:** audit performance; audit opinion shopping; audit expectation gap; internal control weakness; audit report lag; social capital; audit report quality

## 1. Introduction

In the competitive environment of auditing and following big scandals such as Enron Corp, Global Crossing, Arthur Andersen, etc., as well as reviewing the findings of the lawsuits raised in America and England in the late 19th and 20th centuries, the public's attention is drawn to The performance of the auditors and the quality of their reports attracted attention because the public opinion is that the inappropriate level of performance of the auditors and the poor quality of the audit has caused such events to occur; As a result, it is important to pay attention to the performance of auditors and to know the factors affecting it. Auditors' performance has a direct impact on the audit report quality. Accounting principles and auditing standards improve auditors' performance, but unfortunately, sole rules cannot impede ethical issues. Auditors' personality traits are also essential in meeting the users' expectations of financial statements. Identifying the factors that affect the professional ethics of auditors and then implementing a sound plan to improve these factors can help provide a higher quality audit report (Homayoun et al. 2022). Therefore, this study addresses the impact of social capital as a societal asset on auditor performance. The purpose of the audit is to ensure the financial statements prepared by the managers. Therefore, the audited financial statements and audit reports as information sources are concentrated on users. They are the necessary means of communication between auditors and the users of financial statements. Auditors play a significant role in detecting financial fraud and predicting the likelihood of bankruptcy and unexpected crises, and they should provide an unqualified professional opinion to employers (Muñoz-Izquierdo et al. 2019; Lari Dashtbayaz et al. 2022). DeAngelo states that audit quality is caused by the probability of discovering and reporting significant deviations in financial statements. In this

definition, audit quality is a function of the ability to discover important deviations, which is a result of the auditor's competence and ability, and also the auditor's willingness to report these deviations, which is related to the auditor's independence (DeAngelo 1981). Davidson and Neu (1993) define audit quality as an auditor's ability to detect and report material misstatements and detect earning manipulation. The quality of auditors' activities and reports depends on their performance in this field. Baatwah and Al-Ansi (2022), the performance of auditors is caused by various factors, including the assessment of the amount of assigned audit work, the degree of achievement of planned audit activities, the evaluations made by managers and partners of the audit firm, the ability to manage time and cost. The evaluations made from the stakeholders' viewpoints, the audit methods used for the audit, etc., were evaluated. Hence, auditor performance affects the quality of audit reports and investment decisions and prevents any harm to shareholders. Since audit performance is unobservable before or during the audit, various indicators have been introduced in research to show it; This study examines the factors associated with auditor performance; audit opinion shopping, audit expectation gap, audit report lag, and internal control weakness. Audit opinion and detecting and reporting material misstatements greatly depend on auditors' behaviour while performing their duties (Chan et al. 2009; Salehi et al. 2022a, 2022b). In societies with high social capital, honesty, obligations fulfilment, and mutual communication are among the social norms and values (Salehi et al. 2022c, 2022d). Thus, auditors' virtue, consisting of those personality traits that enable auditors to make professional and ethical judgments, is influenced by social capital. Social Capital has been extensively studied in sociology (Coleman 1988), political science (Woolcock 2010), and economics (Guiso et al. 2004) and offers considerable potential for a better understanding of multilevel management and organizational phenomena (Payne et al. 2011). The Legatum Institute defines social capital basics as the strength of personal relationships, social networks, social norms, and civic participation in a country. Therefore, the presence of this factor in society considerably improves the economy. There are rules and regulations for improving auditor performance. This study further examines whether trust and collaboration also will have a positive impact on auditor performance. Social Capital is among the principal assets, yet, its effect on auditor performance has never been studied. The effect of social capital on auditor performance can also significantly influence financial reporting quality.

## 2. Literature Review and Hypotheses Development

### 2.1. Audit Performance

Performance can be measured according to several factors; according to Bonner and Sprinkle (2002), performance can be measured from three aspects of personal characteristics (such as the amount of knowledge, use of knowledge, self-confidence, responsibility, internal motivation, value cultural factors, etc.), characteristics related to the task (such as complexity, processing style, response style, etc.) and characteristics related to environmental conditions (such as time pressure, achieving the set goal, responsibility, etc.) classified (Salehi and Dastanpoor 2021). In auditing institutions, the performance of auditors is affected by different factors. Alissa et al. (2014) consider the audit performance not only influenced by the inherent complexity of the audited company but also by the effect of auditors' responsibilities and duties and auditor characteristics. Many types of research in the audit field also show that audit performance increases with the increase of effort and experience of auditors (Kanfer and Ackerman 1989; Simnett 1996; Yeo and Neal 2004; Lim and Tan 2010) and with the complexity of the work in Audit firms are declining (Tan et al. 2002). Afifah et al. (2015) stated three factors that are effective on the auditor's performance, such as role conflict, self-efficacy, and sensitivity to the auditor's professional ethics, and found that the role conflict is due to the mismatch between the auditor's expectations in the organization and the expectations of other people inside or outside. Organization arises, it can cause job dissatisfaction, reduce the motivation of auditors, and negatively affect the auditor's overall performance (Fanani et al. 2008). Also, auditors with a high level of self-

efficacy are highly motivated and always focus all their efforts and on achieving effective performance in their judgments in achieving goals. Also, the sensitivity of auditors towards the observance of professional ethics, which is one of the other aspects of commitment to the profession, makes auditors strive to fulfill their duties and responsibilities and maintain the quality of their work, and as a result, improve decisions and professional performance (Afifah et al. 2015). Therefore, since the performance of auditors depends on the factors and characteristics of the audited company as well as the characteristics of the auditing institution, in this research, we examine the factors of audit opinion purchase, the audit expectation gap, the weakness of internal controls and the delay of the audit report to examine the performance of auditors as follows:

### 2.2. Audit Opinion Shopping

Audit reporting is among the necessary tools to ensure the reliability of financial statements and other corporate information. The presentation of financial statements is likely to be biased. Therefore, the oversight of independent auditors enables them to reduce such biases. Auditing financial statements by independent auditors is one of the mandatory rules. However, managers play a significant role in recruiting and switching auditors. Managers can replace previous auditors with auditors who issue an opinion favouring management to promote their interests (Lennox 2000; Moradi et al. 2022; Seifzadeh et al. 2021). Auditor rotation has gained increasing attention in recent years. The U.S. Department of Treasury (2008) states that auditor switches are intensely growing, and there is still no obligation to disclose the reason for the change. Johnson and Lys (1995), Lennox (2000), Woo and Koh (2001), and Hudaib and Cooke (2005) have concluded that issuing a qualified report increases the probability of an auditor's switch. It is assumed that companies will change their auditor after receiving the modified audit report, which reduces audit quality (Khaksar et al. 2022). This issue will impede auditor independence and power, and negative consequences will follow. Audit opinion shopping refers to switching auditors by audit clients to obtain a clean audit opinion from the successor auditor. Jha and Chen (2015) have defined opinion shopping as a practice by audit clients to look for auditors willing to present a more favourable audit opinion. Lennox and Pratt (2003) mentioned that audit opinion shopping occurs when the company switches and retains auditors to prevent unfavourable opinions. Securities and Exchange Organization in the U.S. regards opinion shopping as a practice in that auditors help the company achieve its objectives even if the report's credibility is hampered (Archambeault and DeZoort 2001). Opinion shopping is an issue that is hard to measure due to the high incentive to conceal it (Archambeault and DeZoort 2001). Lennox (2000) shows that companies will be successful in opinion shopping. Opinion shopping has posed concerns for regulators over several decades Senate. U.S (1976); SEC (1988); EC (2010). Although this is an important issue, DeFond and Zhang (2014) state that the studies in this area are not promising.

### 2.3. Audit Expectation Gap

Auditing improves the quality of reported financial information and provides economic benefits to the reporting organization and its external members (Wallace 1987). The most fundamental role of independent auditors is to ensure the fairness of financial statements to comply with generally accepted accounting principles and their reliability. Auditors evaluate and judge the information in financial statements and express their professional opinion through an audit report (Lari Dashtbayaz et al. 2019). Although this insurance is not absolute, users of auditing services consider them as a means of establishing assurance about the information provided by management. The standard audit report communicates the auditor's findings (regarding the financial statements) to the users of the audit services. However, communication is a two-way street, and its efficiency depends on the common understanding between the communicator and the addressee regarding the transmitted messages and concepts (Duncan and Moriarty 1998). Regarding the goal of the audit profession, the audit service applicants' perception of the audit objective and the

auditor's responsibilities and plans vary. There is a gap between what financial statement users expect from the audit profession and what auditors have defined as their role in the audit process. This definition is called the expectation gap. American Institute of Certified Public Accountants (AICPA 2011) describes the term as the difference in expectation of users of financial statements and auditors' actual performance. The expectation gap between auditors and the users of audit services has posed considerable concerns for the accounting society and the legislators, so serious attempts have been made to reduce the gap (AICPA 1978). American Institute of Certified Public Accountants (AICPA) 1978 appointed a commission to check if the expectation gap exists. The research showed the gap regarding the audit reports. Hence, some modifications were made to improve the audit report. The results showed a gap in the audit reports. Hence, some modifications were made to improve the audit report. International Auditing and Assurance Standards Board (IAASB) in 2011 and Accounting Standards Board (ASB) in 2010 conducted four research studies to bridge the gap between users' perceptions of audit reports and auditors' understanding of those reports. The results suggested a change in the standard audit report to correctly communicate the audit responsibilities, nature, and effects (Haddrill 2011).

### 2.4. Internal Control Weakness

Today, organizations need a quality management system to guide their organization for increased market competition, development of emerging technologies, the complexity of rules and standards, increased risk of fraud, and increased customer expectations. Designing and implementing an effective internal control system can reduce the risk of the business's goals failing. Internal control is a process designed and established by the organization's board of directors, management, and other employees to ensure the reasonable achievement of the entity's goals (COSO 2013). Internal controls have long been the focus of legislators. Legislators intended to codify the laws and standards to improve the current situation, especially after the global financial scandals, including Enron, WorldCom, and Parmalat. The Sarbanes Oxley Act was passed in 2002, and Section 404 of the Act required companies to publish internal control reports and shifted the focus of management to implementing internal controls. Subsequently, the Tehran Stock Exchange 2012 set guidelines for applying internal controls in Iran. According to Article 12 of this guideline, all companies must establish and implement appropriate internal controls to achieve the firm's objectives. The directors should review the internal controls system annually and disclose its results as the Internal Control Report. If there is a significant weakness in the internal control system that the board of directors has not disclosed, the auditor should state this weakness and its negative consequences. The companies' internal control quality affects the financial reporting quality, auditor decisions, and the reduction of governance problems. Extensive research on internal controls shows that weaknesses in the internal control system reduce the quality of financial reporting (Lin et al. 2011; Doyle et al. 2007; Iliev 2010; Yazawa 2010). According to a study by Ji et al. (2017), designing an effective internal control system ensures financial reporting quality. There is a direct relationship between the quality of internal control and the financial reporting quality that improves audit quality. Hence, business enterprises need an effective internal control system for the following reasons. Such reasons include achieving their short-term and long-term goals, maintaining their financial situation and profitability, surviving accidents, and accountability (COSO 2013).

### 2.5. Audit Report Lag

The nature of the information required to be available on time or its value will be lost; this feature is highly notable in audit reports. Because the delay in providing information affects the decision-making of users as well as the effectiveness of financial statements. Studies indicate a strong market response to timely audited financial statements (Chambers and Penman 1984). The timeliness of audit reports helps the users make the right decisions about audit reports. Audit report lag refers to the time between the end of the firm's fiscal

year and the date that the audit report is issued (Ashton et al. 1987; Knechel and Payne 2001; Bronson et al. 2011; Krishnan and Yang 2009; Whitworth and Lambert 2014) and since the financial statements cannot be published before the audit process is completed, it has been investigated in many studies due to its importance. The findings support that this delay is influenced by the characteristics of the company such as the size of the industry, the presence of accruals, the quality of internal controls (Ashton et al. 1987; Abdillah et al. 2019; Gontara et al. 2022), the characteristics of managers and board members such as gender, financial expertise, ability of managers, turnover of managers, religion (Harjoto et al. 2015; Kalelkar and Khan 2016; Krishnan and Wang 2015; Oradi 2021; Al-Ebel et al. 2020) Audit firm characteristics such as profitability, effectiveness, scope of audit work (complexity), audit staff experience, auditor tenure, auditors' motivation to provide timely reports, non-audit services, fees received by audit institutions (Bamber et al. 1993; Abdillah et al. 2019; Rusmin and Evans 2017; Habib et al. 2019; Lai 2023; Li et al. 2022) and other things such as the tone of annual reports (Teng and Han 2023), cultural dimensions (Toumi et al. 2022), the report is under the ifrs standard (Zhou et al. 2022). Therefore, all these cases will cause audit risk that auditors will spend different time completing the audit work when faced with these risks.

### 2.6. Social Capital

Social Capital refers to the quality of human relations and their correlation with common societal values, including trust, participation, consensus, and empathy. Social Capital facilitates economic growth in society (Putnam 1993; Shleifer et al. 1997; Knack and Keefer 1997). Hanifan first proposed social capital in 1916 in the U.S. Hanifan (1916) called social capital an intangible capital encompassing most people's lives. Such affairs are friendship, sympathy, and all human social interactions. One institution that measures and ranks social capital on the international scale annually is the Legatum Institute. This institute defines social capital foundations as the strength of the following factors. Such factors are personal relationships, social networks, social norms, and civil partnerships in a country. In 2020 among 167 countries, Denmark, Norway, and Finland were ranked as the countries with the highest social capital. Iran also ranked 62, 65, 60, 68, 88, 83, and 70 from 2012 to 2018.

### 2.7. Relationship between Social Capital and Audit Opinion Shopping

Hoi et al. (2019) analyzed the impact of social capital on the firms' management opportunism in providing CEO rewards. The results conveyed lower opportunism in reward calculation and regulation in the companies with higher social capital. Hartlieb et al. (2019) showed that management authority controls the firm's cost and social capital prevents management from opportunistic decision-making concerning cost stickiness. Generally, social capital reduces the opportunistic behaviour of managers and employees. It also increases social participation. Audit opinion shopping increases after the manager's opportunism in issuing the modified audit report. Hence, increasing social capital reduces the chance of opinion shopping. Du et al. (2022) investigated the effect of CEO-auditor dialect connectedness (CADC) on audit opinion purchase (AOS), and their findings showed that the probability of audit opinion purchase for companies with CEO-auditor dialect connectedness is significantly higher. It is more than their counterparts. The finding shows that CADC impairs the auditor's independence.

**H1:** *there is a negative and significant relationship between social capital and audit opinion shopping.*

### 2.8. Relationship between Social Capital and Audit Expectation Gap

A society with high social capital will have higher trust, participation, and sympathy (Salehi and Dastanpoor 2021). It has been mentioned that some of the audit expectation gaps are due to unreasonable public expectations from the audit profession and the deficient performance of auditors. It can be concluded that the expectation gap results from (A) unreasonable expectations of financial statement users and (B) deficient auditor perfor-

mance. Increasing participation, sympathy, and legality will eliminate the expectation gap. Hence, increased social capital will negatively and significantly affect the audit expectation gap. The research evidence of Gao et al. (2021) shows that in societies with high social capital and strong social norms, the unethical behavior of companies is limited and causes more efficient use of company resources. Therefore, it can be expected that companies with a high amount of social capital use will provide more favorable behavior and, as a result, better quality reports, and it is argued that we will see a reduction in the audit expectations gap.

**H2:** *There is a negative and significant relationship between social capital and the audit expectation gap.*

### 2.9. Relationship between Social Capital and Internal Control Weakness

Numerous studies indicate that social capital improves collaboration and social participation. Environments with higher social capital have higher social control (Coleman 1988; Putnam 1993; Fukuyama 1995; Buonanno et al. 2009). Social Capital also matters in a company's environment, improves firm participation and economic enterprise integration, and promotes organizations (Jha and Chen 2015). Shu et al. (2018) analyzed the relationship between organization integration and improving internal controls. They found that integration significantly impacts internal control quality by reducing internal control deficiencies. Hence, increased social capital improves economic integration and minimizes internal control weaknesses. Y. Chen et al. (2016) found that ICW is low for companies with longer auditor tenure, and auditor rotation and auditors' distance from their clients cause auditors to be deprived of their employer's specific knowledge.

**H3:** *There is a negative and significant relationship between social capital and internal control weaknesses*

### 2.10. Relationship between Social Capital and Audit Report Lag

Previous studies indicate that social capital improves human relation quality, reduces opportunistic behaviour, and enhances social participation (Coleman 1988; Putnam 1993; Fukuyama 1995; Buonanno et al. 2009). Social capital benefits society by emphasizing values and ideas that contribute to collaboration (Guiso et al. 2011). Oussii and Taktak (2018) analyzed the impact of coordination between the internal and independent auditor performance concerning audit reports' timeliness and found the following relationship. This relationship reveals that the extent of the coordination between internal and external auditors highly affects audit report lag. Therefore, the higher the coordination, the lower the audit report lag. Enhancing social capital improves human relation quality and organization participation. Social Capital also improves coordination between internal and external auditors leading to a decrease in audit report lag. The analyzes of Daoust and Malsch (2020) show that the effectiveness of the power of auditees is realized from two main sources of power developed over time and in companies, which include specialized knowledge of auditing techniques and social capital. The findings state that the auditees, relying on their cognitive power, use three different power strategies to limit the operational independence of the auditors. On the other hand, auditors may be aware of the risks created by the expertise of the auditee and their social capital. By focusing on it, they interpret the pressures exerted by clients on auditors as a product of strategic actions and discuss the primary consequences of independence risk. Abdullatif et al. (2023) showed a positive and significant relationship between ARL and each audit fee factor, the audit firm's size, and the issuance of an audit opinion.

**H4:** *There is a negative and significant relationship between social capital and audit report lag.*

### 3. Sample Selection, Research Design, and Variable Measurement

*3.1. Sample Selection*

This was applied research that adopted a descriptive-correlational design to pursue its goals. After collecting data from Rahavard Novin Software and the Tehran securities and exchange organization, the data were prepared and organized using Excel software. The primary data and information of this research were used to test the hypotheses using the Kodal system (manual collection of information and research variables from financial statements and reports of the board of directors and managers) and the new Havard data processing software. Then, the Stata software was used to analyze the raw data. This article used the multivariate linear regression model to test the hypothesis, and descriptive and inferential statistical methods were used to analyze the obtained data. Therefore, the descriptive statistics table is used to describe the data. At the inferential level, the Ramsy RESET test, F-Limer test, Hausman test, Wooldridge test and Breusch-Pagan test multivariate linear regression model are used to test the hypothesis. The analyzed sample included 128 Iranian listed firms on the Tehran Stock Exchange from 2014 to 2020. The main reason for investigation in this period was data availability. A systematic deletion method was used to determine the sample size based on the following criteria: 1. The information needed to calculate the research variables for sample companies should be available during the research period. 2. The fiscal year of the firm ends in March. 3. The firm has provided all required information to determine research variables. 4. There has been no change in the fiscal year during the research period. 5. All investment firms, banks, and insurance companies were excluded.

*3.2. Research Design*

**H1** Model:

$$
\begin{aligned}
Shop_{it} = a_0 + \ & a_1 LnSC_{it} + a_2 AChange_{it} + a_3 ATenure_{it} + a_4 LnAFee_{it} + a_5 Big_{it} + a_6 HHI_{it} + a_7 LnSize_{it} \\
& + a_8 Lev_{it} + a_9 ROA_{it} + a_{10} Age_{it} + a_{11} Loss_{it} + a_{12} Ret_{it} + a_{13} Current_{it} + \alpha_{14} AFA + \alpha_{15} Inve \\
& + \varepsilon_{it}
\end{aligned}
$$

The regression coefficient in this model is *a*1. If the coefficient is significant, the first hypothesis (H1) regarding a significant relationship between social capital and audit opinion shopping is confirmed.

**H2** Model:

$$
\begin{aligned}
AEG_{it} = a_0 + \ & a_1 LnSC_{it} + a_2 Achange_{it} + a_3 ATenure_{it} + a_4 LnAFee_{it} + a_5 Big_{it} + a_6 HHI_{it} + a_7 LnSize_{it} \\
& + a_8 Lev_{it} + a_9 ROA_{it} + a_{10} Age_{it} + a_{11} Loss_{it} + a_{12} Ret_{it} + a_{13} Current_{it} + \alpha_{14} AIS + \alpha_{15} AQ \\
& + \alpha_{16} SChange + \varepsilon_{it}
\end{aligned}
$$

The regression coefficient in this model is *a*1. The main hypothesis (H2) regarding a significant relationship between social capital and the audit expectation gap is confirmed if the coefficient is significant.

**H3** Model:

$$
\begin{aligned}
ICW_{it} = a_0 + \ & a_1 LnSC_{it} + a_2 MTenure_{it} + a_3 MChange_{it} + a_4 MSI_{it} + a_5 MSF_{it} + a_6 BSI_{it} + a_7 BSF_{it} + a_8 Blnd_{it} \\
& + a_9 SalesGrowth_{it} + a_{10} Inteng/TAsstes_{it} + a_{11} LnSize_{it} + a_{12} Lev_{it} + a_{13} ROA_{it} + a_{14} Age_{it} \\
& + \varepsilon_{it}
\end{aligned}
$$

The regression coefficient in this model is *a*1. The main hypothesis (H3) regarding a significant relationship between social capital and internal control weakness is confirmed if the coefficient is significant.

**H4** Model:

$$
\begin{aligned}
Delay_{it} = a_0 + \ & a_1 LnSC_{it} + a_2 Achange_{it} + a_3 ATenure_{it} + a_4 LnAFee_{it} + a_5 Big_{it} + a_6 HHI_{it} + a_7 LnSize_{it} \\
& + a_8 Lev_{it} + a_9 ROA_{it} + a_{10} Age_{it} + a_{11} Loss_{it} + a_{12} Ret_{it} + a_{13} Current_{it} + \alpha_{14} AQ + \alpha_{15} Risk \\
& + \alpha_{16} AFA + \varepsilon_{it}
\end{aligned}
$$

The regression coefficient in this model is *a*1. The main hypothesis (H4) regarding a significant relationship between social capital and audit report lag is confirmed if the coefficient is significant. In the following, we will define each model variable.

*3.3. Variable Measurement*

3.3.1. Dependent Variable

***Audit opinion shopping:*** this dummy variable is estimated according to previous studies (F. Chen et al. 2016; Lennox 2000). Suppose financial statements are re-stated, and audit reports are unqualified, but the client has not changed the auditor or has substituted the auditor with a less qualified one. In that case, it equals one, and otherwise, zero. Here quality means that the client has substituted the auditor with a lower quality audit firm. To have a more stringent criterion for evaluating audit opinion shopping, we added the following factors to the above ones. If the client has switched auditors in the following situations, opinion shopping equals one and otherwise zero. The client has changed auditor to an equal quality audit (the same audit rank) but less audit fee with re-stated financial statements or when the client has changed the specialization industry auditor with a non-specialized industry auditor.

***Audit expectation gap:*** This term usually describes a difference in the expectations of specialists (auditors) and another group that relies on this speciality (financial statements' users). Regarding auditor responsibility, public perception differs from the audit profession. It is an audit expectation gap (Ruhnke and Schmidt 2014). The AEG is estimated by the absolute error of Salehi et al. (2019).

Model 1:

To examine the expectation gap, first, we calculate the absolute value of stock price changes using the contributing factors based on the following equation:

$$
\begin{aligned}
|ASP|_{it} = \beta_0 + \ & \beta_1 profit\ and\ loss_{it} + \beta_2 industry_{it} + \beta_3 change\ board_{it} + \beta_4 inflation_{it} \\
& + \beta_5 earning\ persistence_{it} + \beta_6 price\ earnings\ ratio_{it} + \beta_7 the\ liquidity_{it} + \beta_8 debt\ ratio_{it} \\
& + \beta_9 dividends\ per\ share_{it} + \beta_{10} capital\ structure_{it} + \beta_{11} capital\ increase_{it} \\
& + \beta_{12} forecast\ earnings\ per\ share_{it} + \beta_{13} turnover_{it} + \beta_{14} return\ on\ assets_{it} \\
& + \beta_{15} stock\ returns_{it} + \beta_{16} exchange\ rate_{it} + \beta_{17} oil\ price_{it} + \beta_{18} election_{it} \\
& + \beta_{19} current\ ratio_{it} + \beta_{21} quick\ ratio_{it} + \varepsilon_{it}
\end{aligned}
$$

$|ASP|$: The absolute value of stock price changes three days before and after disclosing financial statements and audit reports.

*Profit and loss*: is an indicator variable equal to 1 if the firm reports profit; otherwise, 0.

The official industry categories established by the Tehran stock exchange market are used.

Change board is an indicator variable equal to 1 if at least one board member has changed and 0 otherwise.

Inflation: the fluctuation of the inflation rate and values are extracted from the quarterly reports of the Iranian central bank.

Earning persistence: earning persistence is calculated with inverse errors of Model 2, as follows.

Model 2:

$$EARN_{i,t} = \alpha_0 + \alpha_1 EARN_{i,t-1} + \varepsilon_{it}$$

$EARN_{it}$: earnings of the current year.

$EARN_{i,t-1}$: the earnings of the previous year.

Price-earnings ratio: stock price divided by earnings per share (P/E).

The liquidity: is stock liquidity, which is calculated as follows:

$$BAS = \frac{AP - BP}{\frac{AP+BP}{2}} * 100$$

*BAS*: the difference between purchasing and selling stocks price of the firm.

*AP*: average proposed price for selling firm stocks.

*BP*: average proposed price for purchasing firm stocks.

Debt ratio: the proportion of total debts to total assets.

Dividend per share: The sum of declared dividends issued by a company for every ordinary share outstanding is divided into the firm's total shares.

Capital structure: capital structure is calculated with the following equation:

$$ML_{it} = \frac{BD_{it}}{BD_{it} + ME_{it}}$$

$ML_{it}$: financial leverage based on the market value of company *i* in year *t*.

$BD_{it}$: book value of debts for the company *i* in year *t*.

$ME_{it}$: market value of equity for the company *i* in year *t* (market value of equity is computed through the market value of shares multiplied by its number).

The capital increase is an indicator variable equal to 1 if the firm experiences a capital increase and 0 otherwise.

Forecast earnings per share: is an indicator variable that equals 1 if the real earnings of the company *i* in year *t* are more than the forecasted earnings, and 0 otherwise.

Turnover: the number of transacted shares of the company *i* in year *t* is considered as turnover.

Model 3 errors are used to the extent possible to control the share price effects.

Model 3:

$$VOL_{it} = \beta_0 + \beta_1 MVOL_t + \varepsilon_{it}$$

$$MVOL_t = \frac{No : of\ transacted\ shares\ in\ the\ entire\ market}{No : of\ published\ shares\ in\ the\ market}$$

$$VOL_{it} = \frac{No : of\ transacted\ shares\ in\ the\ company\ i}{No : of\ published\ shares\ in\ the\ company\ t}$$

Return on assets: net profit divided by mean total assets.

Stock return: is calculated with the following equation:

$$R = \frac{(base\ price - day\ price\ ) + DPS + right\ issues + compensation\ price}{base\ price + (1000 * percentage\ of\ capital\ increase\ from\ input)} * 100$$

Exchange rate: currency change, which is extracted from the central bank.

Oil prices: the average price of oil in year t.

The election is an indicator variable equal to 1 if there is a presidential election in year t and 0 otherwise.

Current ratio: the current assets divided by the current debt.

Quick ratio: current assets minus inventories divided by current debts

*Internal control weakness:* The significant weakness of internal controls in external auditors' reports is used to evaluate the weakness. According to Munsif et al. (2012), it equals one and otherwise zero. The following items are used to examine the relationship between social capital (an independent variable) and internal controls' weakness (a dependent variable). These items are social capital impact on financial and non-financial internal controls deficiencies and IT internal controls weakness. Therefore, 800 reports from external auditors were examined and the data on internal control weaknesses were extracted.

*Audit Report lag:* this variable is the time between the balance sheet date and when auditing financial statements are finished (Carslaw and Kaplan 1991; Bamber et al. 1993).

### 3.3.2. Independent Variables

*Social capital:* this variable is obtained from the Log of Iran's social capital score in the Legatum ranking.

### 3.3.3. Control Variables

The control variables of the research along with how to measure each one are as described in the Table 1.

**Table 1.** The Control variables.

| Variable | Description | Source |
|---|---|---|
| Audit Change | If the auditor changes in the year under review, it equals one; otherwise zero. | Bronson et al. (2011); Munsif et al. (2012); Oussii and Taktak (2018); Khaksar et al. (2022) |
| Audit Tenure | The period that the auditor continuously audited business until the year under review | F. Chen et al. (2016); Lari Dashtbayaz et al. (2019); Khaksar et al. (2022) |
| Ln Audit Fee | Natural log of audit fee | Jha and Chen (2015) |
| Audit Firm size | If the audit firm is an Iranian Auditing Organization, Mofid Rahbar or any audit firm that mandatory auditor rotation is not required, it equals one and otherwise zero. | F. Chen et al. (2016) |
| Herfindahl–Hirschman Index | $\left(\frac{received\ audit\ fees}{total\ industry\ audit\ fees}\right)^2$ | Habib and Bhuiyan (2011); Munsif et al. (2012) |
| Ln Firm Size | Natural logarithm of firms' total assets | Archambeault and DeZoort (2001); Carslaw and Kaplan (1991); F. Chen et al. (2016); Ji et al. (2017); Oussii and Taktak (2018); Hoi et al. (2019) |
| Financial Leverage | equals the ratio of total debts to total assets | Archambeault and DeZoort (2001); Carslaw and Kaplan (1991); F. Chen et al. (2016) |
| Return on Assets | equals the ratio of net profit divided by the total assets | Oradi (2021) |
| Firm Age | equals to the time since the firm was established to the year under review | F. Chen et al. (2016) |
| Report Loss | if the firm is at a loss during the year under review, it equals one; otherwise, zero. | F. Chen et al. (2016) |
| Total Stock Return | $\frac{dividends+(initial\ stock\ price-ending\ stock\ price\ \text{period 1})}{initial\ stock\ price}$ | Hoi et al. (2019); Seifzadeh et al. (2021) |
| Current Ratio | equals the ratio of current assets divided by current liability | Oradi (2021) |
| Management Tenure | the period the manager was in charge permanently until the year under review. | Hoi et al. (2019) |
| Management Change | If the manager was changed in the year under review, it equals one and otherwise, zero | Woo and Koh (2001); Chan et al. (2009); Hartlieb et al. (2019) |
| Management industry specialization | If the manager has a degree relevant to the mentioned industry, it equals one; otherwise zero. | Salehi et al. (2021b) |
| Management Financial Specialization | if the manager has a degree relevant to one of the financial majors, it equals one; otherwise, zero. | Baatwah et al. (2015); Alzeban (2020) |
| Boards Industry specialization | if at least one board member has a relevant degree in the mentioned industry, it equals one; otherwise, zero. | Salehi et al. (2021b) |
| Board financial specialization | if at least one board member has a relevant degree in one of the financial fields, it equals one; otherwise zero. | Salehi et al. (2021b) |
| Board independence | the ratio of irresponsible board members to the total board member | Hartlieb et al. (2019) |
| The ratio of Fixed Assets to Net Worth | The ratio of fixed assets to total assets | Cohen and Leventis (2013) |
| Sales Growth | This year's sales minus last year's sales, divided by the last year's sales | F. Chen et al. (2016) |
| Audit firm age | From the time since the audit firm was founded to the present year | F. Chen et al. (2016); Du et al. (2022) |
| Institutional Investors | The share percentage of institutional companies' ownership | Hoi et al. (2019); Seifzadeh et al. (2021); Du et al. (2022) |

**Table 1.** *Cont.*

| Variable | Description | Source |
|---|---|---|
| Auditor industry specialization | the market share is used as an index for auditor specialization in the industry<br>The more the market share, the higher is auditor industry specialization and experience.<br>The market share is calculated as below:<br>$\frac{total\ assets\ of\ all\ the\ clients\ in\ every\ specialized\ audit\ firm\ in\ an\ specialized\ indusrty}{total\ assets\ of\ all\ the\ clients\ in\ an\ specialized\ industry}$<br>In this study, the industry specialized firm has a market share more than the above ratio [1.2 ∗ (the number of the companies)]<br>If the result of the above equation is greater than 1.2 (the number of companies), the specialization in the industry of audit firms is confirmed. So the specialized firms equal one, and other firms equal zero. | F. Chen et al. (2016); Khaksar et al. (2022) |
| Audit quality | Audit firm size, auditor tenure, and auditor industry specialization are proxies for audit quality.<br>The audit firm size: if the firm's auditor is an Iranian auditing organization, it equals one; otherwise, zero.<br>Auditor tenure: if the audit firm has tenured for four years or more, it equals one; otherwise, zero.<br>Auditor industry specialization: if an audit firm is industry specialized, it equals one; otherwise zero. | Carslaw and Kaplan (1991); Khaksar et al. (2022); Salehi et al. (2021a) |
| Change the standard | According to changes and revisions in standards, if the year under review is the year with these changes or the following year, it equals one; otherwise zero. | Bronson et al. (2011) |
| Company Risk | total accounts receivable plus inventories divided by the company's total assets in the year under review | F. Chen et al. (2016); Jha and Chen (2015) |

## 4. Research Findings

### 4.1. Descriptive Statistics

In Table 2, the descriptive statistics related to the quantitative and qualitative variables of the research, including the mean, standard deviation, minimum and maximum values, are presented as follows.

**Table 2.** The Descriptive statistic of variables.

| Variable Type | Variable | Symbol | Number of Samples | Min | Mean | Max | SD |
|---|---|---|---|---|---|---|---|
| Quantitative | Audit Expectation Gap | AEG | 800 | 0.000 | 0.180 | 1.936 | 0.219 |
| | Audit Report Delay | Delay | 800 | 2.890 | 4.234 | 4.905 | 0.385 |
| | Ln Social Capital | LnSC | | 3.908 | 3.954 | 3.973 | 0.022 |
| | Audit Tenure | ATenure | 800 | 1 | 3.02 | 13 | 2.747 |
| | Ln Audit Fee | LnAFee | 800 | 2.302 | 7.356 | 14.390 | 1.631 |
| | Herfindahl-Hirschman Index | HHI | 800 | 0.019 | 0.233 | 1 | 0.223 |
| | Financial Leverage | Lev | 800 | 0.061 | 0.561 | 1.010 | 0.179 |
| | Ln Audit Firm Size | Ln Size | 800 | 2.354 | 2.651 | 2.963 | 0.102 |
| | Firm Age | Age | 800 | 10 | 38.693 | 67 | 13.337 |
| | Return on Assets | ROA | 800 | −0.297 | 0.130 | 1.241 | 0.147 |
| | Current Ratio | Current | 800 | 0.270 | 1.581 | 13.150 | 1.115 |
| | Return on Stock | Ret | 800 | −64.485 | 54.587 | 859.498 | 103.072 |
| | Audit firm age | AFA | 800 | 2 | 15.268 | 31 | 7.010 |
| | institutional investors | Inve | 800 | 0 | 0.259 | 0.934 | 0.162 |
| | Audit Quality | AQ | 800 | 0.000 | 0.091 | 1.033 | 0.103 |
| | Management Tenure | MTenure | 800 | 1 | 3.667 | 15 | 3.089 |
| | Board independence | BInd | 800 | 0 | 0.703 | 1 | 0.186 |
| | Sales Growth | GrowthSales | 800 | −0.825 | 1.584 | 902.671 | 32.517 |
| | The ratio of Fixed Asset to total assets | Inteng/TAsstes | 800 | 0 | 0.007 | 1.465 | 0.052 |
| | Company Risk | Risk | 800 | 0.003 | 0.579 | 11.301 | 0.537 |

**Table 2.** *Cont.*

| Variable Type | Variable | Symbol | Number of Samples | Min | Mean | Max | SD |
|---|---|---|---|---|---|---|---|
| Qualitative | Audit Opinion Shopping | Shop | 800 | 0 | 0.358 | 1 | 0.479 |
| | Financial Internal Control Weakness | ICWF | 800 | 0 | 0.712 | 1 | 0.452 |
| | Non-financial Internal Control Weakness | ICWOF | 800 | 0 | 0.936 | 1 | 0.244 |
| | IT Internal Control Weakness | ICWIT | 800 | 0 | 0.151 | 1 | 0.358 |
| | Audit Change | AChange | 800 | 0 | 0.381 | 1 | 0.485 |
| | Size of the Auditing Firm | Big | 800 | 0 | 0.281 | 1 | 0.449 |
| | Report Loss | Loss | 800 | 0 | 0.08 | 1 | 0.271 |
| | Auditor industry specialization | AIS | 800 | 0 | 0.437 | 1 | 0.496 |
| | Change the standard | SChange | 800 | 0 | 0.415 | 1 | 0.493 |
| | Management Change | MChange | 800 | 0 | 0.281 | 1 | 0.449 |
| | Management industry specialization | MSI | 800 | 0 | 0.274 | 1 | 0.446 |
| | Management Financial Specialization | MSF | 800 | 0 | 0.419 | 1 | 0.493 |
| | Boards Industry specialization | BSI | 800 | 0 | 0.932 | 1 | 0.251 |
| | Boards Financial specialization | BSF | 800 | 0 | 0.893 | 1 | 0.308 |

*4.2. Test Results*

4.2.1. First Hypothesis

The results for H1 showed that it is a panel model with fixed effects. Therefore, testing the first hypothesis using the fixed effects method is reported in Table 3. Given the sign of the obtained coefficient of regression ($-0.304$), it can be concluded that social capital has a negative effect on audit opinion shopping (t = $-5.33$, Sig. < 0.01). It shows that audit opinion shopping decreases with an increase in social capital. Per control variables, it can be seen that social capital has a significant positive relationship with audit fee, Herfindahl-Hirschman index, financial leverage, report loss, and current ratio and has a significant negative relationship with audit change, audit tenure, return on stock and audit firm age But there is no significant relationship between social capital and the rest of the variables. According to the statistics, the model implies that explanatory variables in the model explained about 6% of the variance in the dependent variable ($R^2$ = 5.7%), and the estimated model was generally significant (F = 2.63, Sig. < 0.01).

**Table 3.** The Results of testing the first hypothesis.

$$Shop_{it} = a_0 + a_1 LnSC_{it} + a_2 AChange_{it} + a_3 ATenure_{it} + a_4 LnAFee_{it} + a_5 Big_{it} + a_6 HHI_{it} + a_7 LnSize_{it} + a_8 Lev_{it}$$
$$+ a_9 ROA_{it} + a_{10} Age_{it} + a_{11} Loss_{it} + a_{12} Ret_{it} + a_{13} Current_{it} + \alpha_{14} AFA + \alpha_{15} Inve + \varepsilon_{it}$$

| Explanatory Variable | Symbol | VIF | Regression Coefficient | t Statistics | Sig. |
|---|---|---|---|---|---|
| Ln Social Capital | LnSC | 1.05 | $-0.304$ | $-5.330$ | 0.000 * |
| Audit Change | AChange | 1.56 | $-0.048$ | $-2.610$ | 0.009 * |
| Audit Tenure | ATenure | 2.50 | $-0.036$ | $-3.170$ | 0.002 * |
| Ln Audit Fee | LnAFee | 1.05 | 0.084 | 3.120 | 0.002 * |
| Size of the Auditing Firm | Big | 2.28 | 0.097 | 1.330 | 0.185 |
| Herfindahl-Hirschman Index | HHI | 1.18 | 0.273 | 5.620 | 0.000 * |
| Ln Firm Size | Ln Size | 1.36 | $-0.964$ | $-1.160$ | 0.248 |
| Financial Leverage | Lev | 2.03 | 0.341 | 1.740 | 0.082 *** |
| Return on Assets | ROA | 1.72 | $-0.278$ | $-1.360$ | 0.175 |
| Firm Age | Age | 1.08 | $-0.012$ | $-1.200$ | 0.231 |
| Report Loss | Loss | 1.26 | 0.263 | 3.830 | 0.000 * |
| Return on Stock | Ret | 1.08 | $-0.002$ | $-3.310$ | 0.001 * |
| Current Ratio | Current | 1.70 | 0.062 | 2.040 | 0.042 ** |
| Audit firm age | AFA | 2.29 | $-0.006$ | $-2.880$ | 0.004 * |
| institutional investors | Inve | 1.18 | $-0.165$ | $-1.150$ | 0.253 |

| | |
|---|---|
| The determination coefficient ($R^2$) | 0.057 |
| F statistics | 2.630 |
| Level of significance F | 0.000 |
| Ramsy RESET test | 0.450 (0.714) |
| F-Limer test | 5.390 (0.000) |
| Hausman test | 23.210 (0.056) |
| Wooldridge test | 4.132 (0.044) |
| Breusch-Pagan test | 4.360 (0.036) |

Notes: *, **, *** Significant at the 0.01, 0.05 and 0.10 levels, respectively.

4.2.2. Second Hypothesis

The results for H2 showed that it is a panel model with fixed effects. Therefore, testing the second hypothesis using the fixed effects method is reported in Table 4. Given the sign of the obtained coefficient of regression (−0.547), it can be concluded that social capital has a negative effect on the audit expectation gap (t = −6.87, Sig. < 0.01). It shows that the audit expectation gap decreases with increased social capital. By control variables, it showed that social capital has a significant positive relationship with the size of the auditing firm, audit firm size, financial leverage, return on the stock, auditor industry specialization, and audit quality and has a significant negative relationship with audit change, audit tenure, audit fee, firm Age and report Loss But there is no significant relationship between social capital and the rest of the variables. According to the statistics, the model implies that explanatory variables in the model explained about 6% of the variance in the dependent variable ($R^2$ = 6.3%), and the estimated model was generally significant (F = 2.77, Sig. < 0.01).

**Table 4.** The Results of testing the second hypothesis.

$$AEG_{it} = a_0 + a_1 LnSC_{it} + a_2 Achange_{it} + a_3 ATenure_{it} + a_4 LnAFee_{it} + a_5 Big_{it} + a_6 HHI_{it} + a_7 LnSize_{it} + a_8 Lev_{it} \\ + a_9 ROA_{it} + a_{10} Age_{it} + a_{11} Loss_{it} + a_{12} Ret_{it} + a_{13} Current_{it} + \alpha_{14} AIS + \alpha_{15} AQ + \alpha_{16} SChange + \varepsilon_{it}$$

| Explanatory Variable | Symbol | VIF | Regression Coefficient | t Statistics | Sig. |
|---|---|---|---|---|---|
| Ln Social Capital | LnSC | 1.270 | −0.547 | −6.870 | 0.000 * |
| Audit Change | AChange | 1.570 | −0.053 | −2.700 | 0.007 * |
| Audit Tenure | ATenure | 2.190 | −0.018 | −3.170 | 0.002 * |
| Ln Audit Fee | LnAFee | 1.060 | −0.048 | −3.330 | 0.001 * |
| Size of the Auditing Firm | Big | 1.870 | 0.075 | 2.020 | 0.044 ** |
| Herfindahl-Hirschman Index | HHI | 1.160 | −0.060 | −0.860 | 0.391 |
| Ln Firm Size | Ln Size | 1.390 | 0.088 | 2.320 | 0.022 ** |
| Financial Leverage | Lev | 2.020 | 0.018 | 22.250 | 0.000 * |
| Return on Assets | ROA | 1.720 | 0.128 | 1.160 | 0.245 |
| Firm Age | Age | 1.090 | −0.002 | −1.810 | 0.073 *** |
| Report Loss | Loss | 1.270 | −0.035 | −3.100 | 0.002 * |
| Return on Stock | Ret | 1.450 | 0.000 | 2.310 | 0.021 ** |
| Current Ratio | Current | 1.810 | −0.023 | −1.450 | 0.146 |
| Auditor industry specialization | AIS | 1.540 | 0.170 | 2.900 | 0.004 * |
| Audit Quality | AQ | 1.180 | 0.224 | 1.670 | 0.095 *** |
| Change the standard | SChange | 1.590 | 0.012 | 0.690 | 0.493 |
| The determination coefficient ($R^2$) | | | 0.063 | | |
| F statistics | | | 2.770 | | |
| Level of significance F | | | 0.000 | | |
| Ramsy RESET test | | | 0.970 (0.404) | | |
| F-Limer test | | | 2.500 (0.000) | | |
| Hausman test | | | 38.800 (0.000) | | |
| Wooldridge test | | | 11.387 (0.001) | | |
| Breusch-pagan test | | | 279.230 (0.000) | | |

Notes: *, **, *** Significant at the 0.01, 0.05 and 0.10 levels, respectively.

4.2.3. Third Hypothesis

Because the regression model with panel data tests the hypotheses, we used the tests of F-Limer, Hausman, Wooldridge, Ramsey RESET, and Breusch-pagan. The results for H3 showed that it is a panel model with fixed effects.

(a) Therefore, testing the third hypothesis (a) using the fixed effects method is reported in Table 5. Given the sign of the obtained coefficient of regression (−1.310), it can be concluded that social capital has a negative effect on financial internal control weakness (t = −2.06, Sig. < 0.05). It shows that financial internal control weakness decreases with an increase in social capital. Per control variables, it can be seen that social capital has a significant positive relationship with MSI, MSF, BSF, board independence, and audit firm size and has a significant negative relationship with management tenure, management

change, Inteng/TAsstes, financial leverage, return on assets and firm age. But there is no significant relationship between social capital and the rest of the variables. According to the statistics, the model implies that explanatory variables in the model explained about 7% of the variance in the dependent variable ($R^2$ = 7.3%), and the estimated model was generally significant (F = 3.23, Sig. < 0.01).

**Table 5.** The Results of testing the third hypothesis (a).

$$ICWF_{it} = a_0 + a_1 LnSC_{it} + a_2 MTenure_{it} + a_3 MChange_{it} + a_4 MSI_{it} + a_5 MSF_{it} + a_6 BSI_{it} + a_7 BSF_{it} + a_8 Blnd_{it}$$
$$+ a_9 SalesGrowth_{it} + a_{10} Inteng/TAsstes_{it} + a_{11} LnSize_{it} + a_{12} Lev_{it} + a_{13} ROA_{it} + a_{14} Age_{it} + \varepsilon_{it}$$

| Explanatory Variable | Symbol | VIF | Regression Coefficient | t Statistics | Sig. |
|---|---|---|---|---|---|
| Ln Social Capital | LnSC | 1.040 | −1.310 | −2.060 | 0.039 ** |
| Management Tenure | MTenure | 1.420 | −0.018 | −2.340 | 0.020 ** |
| Management Change | MChange | 1.320 | −0.300 | −3.320 | 0.001 * |
| Management industry specialization | MSI | 1.300 | 0.292 | 2.420 | 0.016 ** |
| Management Financial Specialization | MSF | 1.340 | 0.084 | 3.120 | 0.002 * |
| Boards Industry specialization | BSI | 1.040 | −0.086 | −1.070 | 0.287 |
| Boards Financial specialization | BSF | 1.090 | 0.026 | 2.260 | 0.024 ** |
| Board independence | BInd | 1.090 | 0.162 | 2.770 | 0.006 * |
| Sales Growth | GrowthSales | 1.020 | 0.000 | 0.350 | 0.725 |
| The ratio of Fixed Asset to total assets | Inteng/TAsstes | 1.020 | −0.605 | −2.310 | 0.021 ** |
| Ln Firm Size | Ln Size | 1.110 | 0.170 | 3.700 | 0.000 * |
| Financial Leverage | Lev | 1.740 | −0.046 | −3.080 | 0.002 * |
| Return on Assets | ROA | 1.640 | −0.588 | −2.920 | 0.004 * |
| Firm Age | Age | 1.060 | −0.032 | −2.990 | 0.003 * |

| | |
|---|---|
| The determination coefficient ($R^2$) | 0.073 |
| F statistics | 3.230 |
| Level of significance F | 0.000 |
| Ramsy RESET test | 0.850 (0.4647) |
| F-Limer test | 3.620 (0.000) |
| Hausman test | 35.410 (0.000) |
| Wooldridge test | 4.111 (0.045) |
| Breusch-Pagan test | 25.700 (0.000) |

Notes: *, ** Significant at the 0.01, 0.05 and 0.10 levels, respectively.

(b) The results of testing the third hypothesis (b) using the fixed effects method are reported in Table 6. Given the sign of the obtained coefficient of regression (−0.173), it can be concluded that social capital has a negative effect on non-financial internal control weakness (t = −2.85, Sig. < 0.01). It shows that with an increase in social capital, non-financial internal control weakness decreases. By control variables, it can be seen that social capital has a significant positive relationship with the management change, MSI, and Inteng/TAsstes and has a significant negative relationship with management tenure, BSI, board independence, and sales growth, financial leverage, return on assets and firm age. But there is no significant relationship between social capital and the rest of the variables. According to the statistics, the model implies that explanatory variables in the model explained about 4% of the variance in the dependent variable ($R^2$ = 4.4%), and the estimated model was generally significant (F = 1.88, Sig. < 0.05).

**Table 6.** The Results of testing the third hypothesis (b).

$$ICWOF_{it} = a_0 + a_1 LnSC_{it} + a_2 MTenure_{it} + a_3 MChange_{it} + a_4 MSI_{it} + a_5 MSF_{it} + a_6 BSI_{it} + a_7 BSF_{it} + a_8 Blnd_{it}$$
$$+ a_9 SalesGrowth_{it} + a_{10} Integ/TAsstes_{it} + a_{11} LnSize_{it} + a_{12} Lev_{it} + a_{13} ROA_{it} + a_{14} Age_{it} + \varepsilon_{it}$$

| Explanatory Variable | Symbol | VIF | Regression Coefficient | t Statistics | Sig. |
|---|---|---|---|---|---|
| Ln Social Capital | LnSC | 1.040 | −0.173 | −2.850 | 0.004 * |
| Management Tenure | MTenure | 1.420 | −0.001 | −2.400 | 0.016 ** |
| Management Change | MChange | 1.320 | 0.027 | 2.330 | 0.020 ** |
| Management industry specialization | MSI | 1.300 | 0.227 | 2.780 | 0.006 * |
| Management Financial Specialization | MSF | 1.340 | −0.033 | −0.950 | 0.343 |
| Boards Industry specialization | BSI | 1.040 | −0.073 | −3.510 | 0.000 * |
| Boards Financial specialization | BSF | 1.090 | 0.052 | 1.160 | 0.247 |
| Board independence | BInd | 1.090 | −0.047 | −2.010 | 0.044 ** |
| Sales Growth | GrowthSales | 1.020 | −0.031 | −3.170 | 0.002 * |
| The ratio of Fixed Asset to total assets | Integ/TAsstes | 1.020 | 0.464 | 2.820 | 0.005 * |
| Ln Firm Size | Ln Size | 1.110 | 0.844 | 1.540 | 0.123 |
| Financial Leverage | Lev | 1.740 | −0.039 | −2.880 | 0.004 * |
| Return on Assets | ROA | 1.640 | −0.343 | −2.710 | 0.007 * |
| Firm Age | Age | 1.060 | −0.012 | −1.840 | 0.066 *** |

| | |
|---|---|
| The determination coefficient ($R^2$) | 0.044 |
| F statistics | 1.880 |
| Level of significance F | 0.026 |
| Ramsy RESET test | 1.470 (0.220) |
| F-Limer test | 1.960 (0.000) |
| Hausman test | 56.150 (0.000) |
| Wooldridge test | 3.297 (0.072) |
| Breusch-pagan test | 147.66 (0.000) |

Notes: *, **, *** Significant at the 0.01, 0.05 and 0.10 levels, respectively.

(c) The results of testing the third hypothesis (c) using the fixed effects method are reported in Table 7. Given the sign of the obtained coefficient of regression (−0.155), it can be concluded that social capital has a negative effect on IT internal control weakness (t = −2.48, Sig. < 0.05). It shows that IT internal control weakness decreases with increased social capital. Per control variables, it can be seen that social capital has a significant positive relationship with the management change, MSI, MSF, BSF, board independence, and audit firm size and has a significant negative relationship with sales growth and financial leverage, and return on assets. But there is no significant relationship between social capital and the rest of the variables. According to the statistics, the model implies that explanatory variables in the model explained about 4% of the variance in the dependent variable ($R^2$ = 4.4%), and the estimated model was generally significant (F = 1.91, Sig. < 0.05).

**Table 7.** The Results of testing the third hypothesis (c).

$$ICWIT_{it} = a_0 + \quad a_1 LnSC_{it} + a_2 MTenure_{it} + a_3 MChange_{it} + a_4 MSI_{it} + a_5 MSF_{it} + a_6 BSI_{it} + a_7 BSF_{it} + a_8 Blnd_{it}$$
$$+ a_9 SalesGrowth_{it} + a_{10} Inteng/TAsstes_{it} + a_{11} LnSize_{it} + a_{12} Lev_{it} + a_{13} ROA_{it} + a_{14} Age_{it} + \varepsilon_{it}$$

| Explanatory Variable | Symbol | VIF | Regression Coefficient | t Statistics | Sig. |
|---|---|---|---|---|---|
| Ln Social Capital | LnSC | 1.040 | −0.155 | −2.480 | 0.013 ** |
| Management Tenure | MTenure | 1.420 | −0.007 | −1.170 | 0.241 |
| Management Change | MChange | 1.320 | 0.062 | 3.900 | 0.000 * |
| Management industry specialization | MSI | 1.300 | 0.093 | 1.970 | 0.049 ** |
| Management Financial Specialization | MSF | 1.340 | 0.088 | 1.900 | 0.059 *** |
| Boards Industry specialization | BSI | 1.040 | −0.075 | −1.130 | 0.259 |
| Boards Financial specialization | BSF | 1.090 | 0.002 | 2.240 | 0.025 ** |
| Board independence | BInd | 1.090 | 0.029 | 1.780 | 0.076 *** |
| Sales Growth | GrowthSales | 1.020 | −0.066 | −2.120 | 0.034 ** |
| The ratio of Fixed Asset to total assets | Inteng/TAsstes | 1.020 | −0.031 | −0.150 | 0.885 |
| Ln Firm Size | Ln Size | 1.110 | 0.163 | 2.610 | 0.009 * |
| Financial Leverage | Lev | 1.740 | −0.334 | −2.180 | 0.029 ** |
| Return on Assets | ROA | 1.640 | −0.065 | −1.890 | 0.059 *** |
| Firm Age | Age | 1.060 | 0.009 | 1.040 | 0.299 |

| | |
|---|---|
| The determination coefficient ($R^2$) | 0.044 |
| F statistics | 1.910 |
| Level of significance F | 0.022 |
| Ramsy RESET test | 1.230 (0.296) |
| F-Limer test | 3.010 (0.000) |
| Hausmen test | 485.900 (0.000) |
| Wooldridge test | 5.194 (0.024) |
| Breusch-pagan test | 46.620 (0.000) |

Notes: *, **, *** Significant at the 0.01, 0.05 and 0.10 levels, respectively.

### 4.2.4. Fourth Hypothesis

Given the sign of the obtained coefficient of regression (−0.053) according to Table 8, it can be concluded that social capital has a negative effect on audit report lag (z = −3.10, Sig. < 0.01). It shows that audit report lag decreases with an increase in social capital. Following control variables, it can be seen that social capital has a significant positive relationship with Herfindahl-Hirschman Index, firm size, firm age, and audit quality and has a significant negative relationship with audit change, audit fee, size of the auditing firm, return on assets, return on stock and company risk. But there is no significant relationship between social capital and the rest of the variables. According to the statistics, the model implies that explanatory variables in the model explained about 6% of the variance in the dependent variable ($R^2$ = 6.3%), and the estimated model was generally significant (chi2 = 49.52, Sig. < 0.01).

**Table 8.** The Results of testing the fourth hypothesis.

$$Delay_{it} = a_0 + \quad a_1 LnSC_{it} + a_2 Achange_{it} + a_3 ATenure_{it} + a_4 LnAFee_{it} + a_5 Big_{it} + a_6 HHI_{it} + a_7 LnSize_{it} + a_8 Lev_{it}$$
$$+ a_9 ROA_{it} + a_{10} Age_{it} + a_{11} Loss_{it} + a_{12} Ret_{it} + a_{13} Current_{it} + \alpha_{14} AQ + \alpha_{15} Risk + \alpha_{16} AFA + \varepsilon_{it}$$

| Explanatory Variable | Symbol | VIF | Regression Coefficient | z Statistics | Sig. |
|---|---|---|---|---|---|
| Ln Social Capital | LnSC | 1.060 | −0.053 | −3.100 | 0.002 * |
| Audit Change | AChange | 1.560 | −0.039 | −2.880 | 0.004 * |
| Audit Tenure | ATenure | 2.500 | 0.009 | 1.400 | 0.162 |
| Ln Audit Fee | LnAFee | 1.050 | −0.002 | −4.270 | 0.000 * |
| Size of the Auditing Firm | Big | 2.260 | −0.023 | −1.720 | 0.085 *** |
| Herfindahl-Hirschman Index | HHI | 1.160 | 0.190 | 3.240 | 0.001 * |
| Ln Firm Size | Ln Size | 1.250 | 0.422 | 1.710 | 0.087 *** |
| Financial Leverage | Lev | 2.050 | 0.094 | 0.720 | 0.472 |
| Return on Assets | ROA | 1.710 | −0.461 | −3.160 | 0.002 * |
| Firm Age | Age | 1.100 | 0.003 | 1.650 | 0.098 *** |
| Report Loss | Loss | 1.260 | −0.008 | −0.260 | 0.795 |
| Return on Stock | Ret | 1.100 | −0.001 | −1.700 | 0.091 *** |
| Current Ratio | Current | 1.830 | 0.020 | 1.310 | 0.191 |
| Audit Quality | AQ | 1.170 | 0.038 | 6.130 | 0.000 * |
| Company Risk | Risk | 1.060 | −0.020 | −1.670 | 0.095 *** |
| Audit firm age | AFA | 2.300 | −0.000 | −0.400 | 0.693 |

| | |
|---|---|
| The determination coefficient ($R^2$) | 0.063 |
| Wald chi2 statistics | 49.520 |
| Level of significance chi2 | 0.000 |
| Ramsy RESET test | 1.440 (0.229) |
| F-Limer test | 13.460 (0.000) |
| Hausmen test | 19.920 (0.174) |
| Wooldridge test | 6.239 (0.013) |
| Breusch-pagan test | 28.740 (0.000) |

Notes: *, *** Significant at the 0.01, 0.05 and 0.10 levels, respectively.

## 5. Discussion and Conclusions

The present study examines the relationship between social capital and audit performance for 128 companies listed on the Tehran Stock Exchange for 9 years from 2014 to 2020. The present study investigated whether there is a relationship between the level of trust and cooperation, personal relationships, social networks, social norms and civic participation, which are considered social capital criteria by Legatum Institute, and the auditor's performance. In this regard, in order to measure the auditor's performance according to the studied research, opinion purchase, audit expectation gap, delay in audit report and weakness of internal controls that can affect the quality of audit report were used.

The results showed that social capital has a positive effect on auditor performance. Hence increased social capital reduces audit opinion shopping, audit expectation gap, internal control weakness, and audit report lag. Augmenting social capital reduces managers' opportunistic behaviour. Therefore, following the modified audit opinion, opinion shopping will be decreased. Increasing social capital enhances the participation, sympathy, and lawfulness between auditors and audit report users. Therefore, the audit expectation gap will be eliminated for the following reasons. Such reasons are (A) unreasonable expectations of financial statement users and (B) the shortage of auditors' real performance. Increasing social capital increases economic integration and social governance. As a result, the company and internal controls will also improve. Jha (2019) investigated the effect of social capital on financial reports and found that companies headquartered in areas with high social capital in the United States were less likely to commit fraud by misrepresenting financial information. In addition, it found that companies located in areas with high social capital have lower levels of discretionary accruals and more readable annual reports. Improving social capital enhances internal and external auditors' collaboration and reduces audit report lag. Dell et al. (2022) in the field of for-profit service organizations, found that social capital positively affects the financial performance of these companies. In contrast,

this study analyzed the relationship between social capital and the financial performance of non-profit organizations and provided evidence that social capital is positively related to the financial performance of non-profit organizations. Also, the results of this research are in line with the findings of Albawwat (2021), which show that all dimensions of social capital indirectly affect audit quality inputs through tacit knowledge sharing. While the cognitive dimension of social capital positively affects auditors' values, ethics, and attitudes, relational social capital has the strongest effect on auditors' knowledge, skills, and experiences.

Since the relationship between social capital and auditor's performance is confirmed, we recommend the following. Such recommendations include upgrading ethnic values by improving social practices. Therefore, we recommend legislators and decision-makers the following items to increase social capital for all organization members, including managers, staff, and internal and external audit members; the such recommendation is setting educational courses. Therefore, all members will be able to improve social capital, and their psychological climate will be supported. Revealing the impact of social capital on audit quality inputs by focusing on the impact of social capital dimensions and also encouraging to explain strategies to increase audit quality increases awareness about the elements affecting audit performance. In addition, the descriptive findings of this study provide audit institutions with a picture of social capital and audit quality inputs, which may be used to draw a perspective for maintaining and improving audits. Also, in this regard, it is possible to provide new insight for the auditing profession about their training and experience in the criteria and components of social capital and specifically to encourage auditors to try to improve their performance. Also, in line with the research, some limitations prevent the research subject from being carried out more widely, such as the limitations of the research, the unavailability and non-disclosure of some research variables, such as the auditor's fee, the expertise of managers, the board of directors, and managers by stock companies, which causes The number of companies was limited.

**Author Contributions:** Conceptualization, M.M.A. and S.J.; methodology, H.R.R.; software, A.A.-j.; validation, H.R.R., formal analysis, A.A.-j.; investigation, S.J.; resources, S.J.; data curation, M.M.A.; writing—original draft preparation, M.M.A. writing—review and editing, M.M.A.; visualization, M.M.A.; supervision M.M.A.; project admin-istration, M.M.A.; funding acquisition, S.J. All authors have read and agreed to the published version of the manuscript.

**Funding:** This research received no external funding.

**Data Availability Statement:** Data will be available at request.

**Conflicts of Interest:** The authors declare no conflict of interest.

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
