# Peer review of "The Effect of Social Capital on Auditor’s Performance"

_jrfm, doi:10.3390/jrfm16020119_

Round 1

Reviewer 1 Report

The paper is interesting, the idea seems to be original, with some relevant results.

Some observations:

1) lines 17-18 (in the abstract): the authors appreciate that "social capital positivly correlates with auditor performance and audit reports"; I think a complement would be necessary, in the sense that the correlation between the "social capital" and the "audit report" is very vague and difficult to support – the correlation could be done between the social capital and the characteristics of the audit report or between the social capital and the quality of the audit reports.

2) line 28: it is vague to specify that the auditor's performance has effects on the audit report; perhaps on the quality of the audit report or on its characteristics.

3) Line 41: The first paper in which this definition of audit quality appears seems to be not Davidson & Neu (1993), but DeAngelo, L.E. (1981) Auditor Size and Audit Quality. Journal of Accounting and Economics, 3, 183-199.

4) Lines 117-118: It seems that the idea is repeated, it appears above, on lines 114-115.

5) section 2.4. The variable audit report lag makes sense when an auditor audits a single company; otherwise, he must make a planning of the works and ARL depends only on the criteria after which the respective programming was made, without necessarily related to the length of the audit mission.

6) Lines 161-162, the last sentence of section 2.4. It seems to have no connection with the paragraph of which it belongs.

7) Section 2.9. The correlation between the social capital and ARL seems extremely interesting precisely because the order in which the auditors audit the companies could be settled according to personal relationships or other characteristics of the relationship between the auditor and the audited. Maybe it would not be bad to make a mention in this regard.

8) Section 3.1.: The authors present the deleting criteria from the sample and the first one is the listing before 2012: this seems to be an inclusion criterion in the sample and not a deleting one; a small reformulation could solve the problem. It is also important to specify the number of observations generated by the 128 Iranian listed companies, if the sample is not balanced.

9) Section 3.1.1. It does not make sense, as long as there is no section 3.1.2. It is sufficient that the text from 3.1.1. be left in the continuation of the text from 3.1.

10) Section 3.2. After each model, the phrase "the regression coefficient in this model is a1" appears. I think it would be better to write "the regression coefficient in this model is ai". Putting ai instead of a1 would better describe the econometric model.

11) Section 3.2.: It would be useful to make a mention that the definition of variables follows in the text a little further.

12) Section 3.2.: The last paragraph only resumes the idea exposed already 4 times; It can be deleted.

13) Line 311, at the end: refer to model 3, but what follows is model 2.

14) Line 344: Reference is made to model 4, but what follows is the model 3.

15) Lines 365-370: It would be extremely useful, for validating data, to present the number of observations in which such ICWs are mentioned in the audit reports.

16) Tables 1: The first variable is defined in a different way from what is presented to the description in section 2.1.

17) Table 1: The formulation of the size the audit firm: ”if the firm is auditor is an Iranian auditing organization, it equals one, oterwise zero appears when describing the variable”. Do we understand that the auditor's size is measured only by its nationality? It's slightly simplistic.

18) Table 1: description of the Company Risk variable: Stock in hand what does it mean? Maybe inventories?

19) immediately after tables 2: I do not think it is necessary to specify the content of statistical sizes such as average, min, max, deviation; In general, potential readers know these things very well.

20) Discussion and Conclusion: The section is very short - it is almost like an abstract - especially since it has two components. The authors could take over some of the key elements of the study here, both from the perspective of the results, and the methodology and the value added in the specific literature. Also, the study has limits that the authors must include in the conclusions section: for example, the limited number of analyzed companies, maybe the number of observations regarding some variables, etc.

Author Response

Dear Reviewer,

Thank you very much for your comments on the paper, the following issues are fixed in the current version:

1) lines 17-18 (in the abstract): the authors appreciate that "social capital positivly correlates with auditor performance and audit reports"; I think a complement would be necessary, in the sense that the correlation between the "social capital" and the "audit report" is very vague and difficult to support – the correlation could be done between the social capital and the characteristics of the audit report or between the social capital and the quality of the audit reports.

Response: With thanks for your attention, The relevant items were corrected according to the honorable judge

2) line 28: it is vague to specify that the auditor's performance has effects on the audit report; perhaps on the quality of the audit report or on its characteristics.

Response: Thanks for your attention, it has been corrected

3) Line 41: The first paper in which this definition of audit quality appears seems to be not Davidson & Neu (1993), but DeAngelo, L.E. (1981) Auditor Size and Audit Quality. Journal of Accounting and Economics, 3, 183-199.

Response: Thanks for your attention, it has been corrected

4) Lines 117-118: It seems that the idea is repeated, it appears above, on lines 114-115.

Response: Thank you for your review; the relevant comment was corrected

5) section 2.4. The variable audit report lag makes sense when an auditor audits a single company; otherwise, he must make a planning of the works and ARL depends only on the criteria after which the respective programming was made, without necessarily related to the length of the audit mission.

Response: Thank you for your attention and accuracy. corrections were made

6) Lines 161-162, the last sentence of section 2.4. It seems to have no connection with the paragraph of which it belongs.

Response: Thank you for your review, The relevant comment was corrected

7) Section 2.9. The correlation between the social capital and ARL seems extremely interesting precisely because the order in which the auditors audit the companies could be settled according to personal relationships or other characteristics of the relationship between the auditor and the audited. Maybe it would not be bad to make a mention in this regard.

Response: Thank you for your valuable comments, corrections have been made

8) Section 3.1.: The authors present the deleting criteria from the sample and the first one is the listing before 2012: this seems to be an inclusion criterion in the sample and not a deleting one; a small reformulation could solve the problem. It is also important to specify the number of observations generated by the 128 Iranian listed companies, if the sample is not balanced.

Response: Thank you for your valuable opinion; the meaning of the sampling method is this: the sampling method is systematic elimination and the specified criteria should be included in the sample. The first item was modified to: The information needed to calculate the research variables for sample companies should be available during the research period

9) Section 3.1.1. It does not make sense, as long as there is no section 3.1.2. It is sufficient that the text from 3.1.1. be left in the continuation of the text from 3.1.

Response: Thank you for your review; the relevant comment was corrected

10) Section 3.2. After each model, the phrase "the regression coefficient in this model is a1" appears. I think it would be better to write "the regression coefficient in this model is ai". Putting ai instead of a1 would better describe the econometric model.

Response: Thank you for your comment. According to the research Bhuiyan and D’Costa (2020 (and Salehi et al., (2020)and many other researchers, the research model has been designed, and if we put the ai coefficient for the first variable, what coefficient should we consider for the next variables? For this reason, while thanking and respecting your comment, respected research reviewer, the coefficients of the regression model did not change. If it is necessary to change, let me know how to insert the coefficients so that the model can be modified.

Bhuiyan, M.B.U. and D’Costa, M. (2020), "Audit committee ownership and audit report lag: evidence from Australia", International Journal of Accounting & Information Management, Vol. 28 No. 1, pp. 96-125. https://doi.org/10.1108/IJAIM-09-2018-0107

Salehi, M., Jahanbin, F., & Adibian, M. S. (2020). The relationship between audit components and audit expectation gap in listed companies on the Tehran stock exchange. Journal of Financial Reporting and Accounting, 18(1), 199-222.

11) Section 3.2.: It would be useful to make a mention that the definition of variables follows in the text a little further.

Response: Thank you for your valuable comments; corrections have been made

12) Section 3.2.: The last paragraph only resumes the idea exposed already 4 times; It can be deleted.

Response: Thank you for your review; the relevant comment was corrected

13) Line 311, at the end: refer to model 3, but what follows is model 2.

Response: Thank you for your review; the relevant comment was corrected

14) Line 344: Reference is made to model 4, but what follows is the model 3.

Response: Thank you for your review; the relevant comment was corrected

15) Lines 365-370: It would be extremely useful, for validating data, to present the number of observations in which such ICWs are mentioned in the audit reports.

Response: Thank you for your review; the relevant comment was corrected

16) Tables 1: The first variable is defined in a different way from what is presented to the description in section 2.1.

Response: The purpose of this variable is to check whether the company had a change of auditor in the year (for example, 2013) or not. Because in the financial reports and the reports of the board of directors of Iran, only the change of the auditor is mentioned without mentioning the reason for the change. And even a company may have received an acceptable opinion the previous year but changed the auditor in the following year for other reasons.

17) Table 1: The formulation of the size the audit firm: ”if the firm is auditor is an Iranian auditing organization, it equals one, oterwise zero appears when describing the variable”. Do we understand that the auditor's size is measured only by its nationality? It's slightly simplistic.

Response: Due to the fact that companies in Iran are not audited by large international institutions, for this reason, several large institutions in Iran have been considered as large and superior institutions in this research.

18) Table 1: description of the Company Risk variable: Stock in hand what does it mean? Maybe inventories?

Response: Thank you for your attention and accuracy; yes, it is true, it means the amount of inventories.

19) immediately after tables 2: I do not think it is necessary to specify the content of statistical sizes such as average, min, max, deviation; In general, potential readers know these things very well.

 Response: Thanks for your comment, dear referee; the text has been removed and corrected

20) Discussion and Conclusion: The section is very short - it is almost like an abstract - especially since it has two components. The authors could take over some of the key elements of the study here, both from the perspective of the results, and the methodology and the value added in the specific literature. Also, the study has limits that the authors must include in the conclusions section: for example, the limited number of analyzed companies, maybe the number of observations regarding some variables, etc.

Response: Thanks to your valuable insight, we tried to update the conclusion as broad as possible. Added content to the practical suggestions section

Reviewer 2 Report

This research is interesting, however, it still needs some revisions, like any other: The sample included 128 firms on the Tehran Stock Exchange from 2014 to 2020, however, they are not updated enough. it is better if the research can reach 2021, the research results until 2021 are more relevant.

The phenomenon or case regarding the auditor's performance in the introduction has not been explained.

In the Literature Review, it is necessary to explain the theory of auditor's performance

and In developing the hypothesis, it is necessary to explain the theory underlying the moderation hub on the hypothesis.

Hypothesis development should be supported by research results from the last 5 years, research results used in data manuscripts are not updated/past.

Sample selection, Research design, and Variable measures need to be explained Population and Purposive sampling used and can be in the form of a table so that it is more informative.

If using Stata software, the research observation period should be longer.

Table 1. The Control variables need to add a column and explain the source.

Table 2. The Descriptive statistics of variables need to add a column for "n"/ number of samples.

The Discussion section is very small and needs further development and is strengthened by Theory and Research Results that support the Last 5 Years and an explanation of each Hypothesis Testing Result.

Author Response

Dear Reviewer,

Thank you very much for the comments on the paper, further all issues are addressed in the current version as follow:

1) This research is interesting, however, it still needs some revisions, like any other: The sample included 128 firms on the Tehran Stock Exchange from 2014 to 2020, however, they are not updated enough. it is better if the research can reach 2021, the research results until 2021 are more relevant.

Response: It was corrected. Due to the fact that the data was available until 2020 in the time frame of this research, unfortunately it was not possible to add the number of years, and now it is not possible to update due to political problems as well as the Internet in Iran and the high number of research variables and time consuming. Thank you for your comment

2) The phenomenon or case regarding the auditor's performance in the introduction has not been explained.

Response: Thank you for your comment, dear reviewer, the relevant comment was made and the relevant items were taken into account

3) In the Literature Review, it is necessary to explain the theory of auditor's performance

Response: Thanks for your comment, dear reviewer, the relevant comment has been made

4) and In developing the hypothesis, it is necessary to explain the theory underlying the moderation hub on the hypothesis.

Response: Thanks for your comment, dear reviewer, the relevant comment has been made

5) Hypothesis development should be supported by research results from the last 5 years, research results used in data manuscripts are not updated/past.

Response: Thanks for your comment, dear reviewer, the relevant comment has been made

6) Sample selection, Research design, and Variable measures need to be explained Population and Purposive sampling used and can be in the form of a table so that it is more informative.

Response: Thank you for your attention. The comment was made

7) If using Stata software, the research observation period should be longer.

Response: The time period of the research is from 2012 to 2020 and 2018 has been entered by mistake

8) Table 1. The Control variables need to add a column and explain the source.

Response: Thanks for your attention, it has been corrected

9) Table 2. The Descriptive statistics of variables need to add a column for "n"/ number of samples.

Response: Thank you for your attention; your comment has been made

10) The Discussion section is very small and needs further development and is strengthened by Theory and Research Results that support the Last 5 Years and an explanation of each Hypothesis Testing Result.

Response: Thanks to your valuable insight, we tried to update the conclusion as broadly as possible. Added content to the practical suggestions section

Round 2

Reviewer 2 Report

the revision has been done well and deserves to be published